# Constructing temporal regulatory cascades in the context of development and cell differentiation

**Rayan Daou**[1], **Tim Beißbarth**[1], **Edgar Wingender**[1], **Mehmet Gültas**[2,3], **Martin Haubrock**[1]*

**1** Department of Medical Bioinformatics, University Medical Center Göttingen, Goettingen, Niedersachsen, Germany, **2** Breeding Informatics Group, Department of Animal Science, Georg-August University, Goettingen, Niedersachsen, Germany, **3** Center for Integrated Breeding Research (CiBreed), Georg-August University, Goettingen, Niedersachsen, Germany

* martin.haubrock@bioinf.med.uni-goettingen.de

**Data Availability Statement:** All relevant data are within the manuscript and its Supporting Information files.

## Abstract

Cell differentiation is a complex process orchestrated by sets of regulators precisely appearing at certain time points, resulting in regulatory cascades that affect the expression of broader sets of genes, ending up in the formation of different tissues and organ parts. The identification of stage-specific master regulators and the mechanism by which they activate each other is a key to understanding and controlling differentiation, particularly in the fields of tissue regeneration and organoid engineering. Here we present a workflow that combines a comprehensive general regulatory network based on binding site predictions with user-provided temporal gene expression data, to generate a a temporally connected series of stage-specific regulatory networks, which we call a temporal regulatory cascade (TRC). A TRC identifies those regulators that are unique for each time point, resulting in a cascade that shows the emergence of these regulators and regulatory interactions across time. The model was implemented in the form of a user-friendly, visual web-tool, that requires no expert knowledge in programming or statistics, making it directly usable for life scientists. In addition to generating TRCs the tool links multiple interactive visual workflows, in which a user can track and investigate further different regulators, target genes, and interactions, directing the tool along the way into biologically sensible results based on the given dataset. We applied the TRC model on two different expression datasets, one based on experiments conducted on human induced pluripotent stem cells (hiPSCs) undergoing differentiation into mature cardiomyocytes and the other based on the differentiation of H1-derived human neuronal precursor cells. The model was successful in identifying previously known and new potential key regulators, in addition to the particular time points with which these regulators are associated, in cardiac and neural development.

**Funding:** R.D FKZ:81X2300184 DZHK https://dzhk.de The funders had no role in study design, data collection and analysis, decision to publish, or preparation of the manuscript. The work was supported by the DZHK (German Center for Cardiovascular Research; FKZ:81X2300184) where DZHK stands for (Deutsches Zentrum für Herz-Kreislauf-Forschung).

**Competing interests:** The authors have declared that no competing interests exist.

# Introduction

Cell differentiation, the building block of development, is a strong representation of regulatory precision. In stem cell differentiation, a handful of regulators kick off a regulatory mechanism that leads to the activation or repression of other regulators and non-regulatory genes, through consecutive waves, starting processes that are geared towards specification and giving rise to different kinds of cells and tissues [1–5]. The discovery of induced pluripotent stem cells (iPSCs) [6–8], opened the door to a rising number of cell differentiation experiments. Owing to the decreasing prices of RNA-seq, these experiments generated a big and growing number of time series datasets that aim to track a certain process of differentiation by taking snapshots of the gene expression at different time points. These datasets could be further analyzed to obtain a better extensive explanatory model of the regulatory processes and to identify new important regulators that can be manipulated to enhance the process. Deriving as much information as possible from such experiments is a crucial goal in the fields of medical and biological research [9–13], yet there is still a need for computational methods that analyze such unique models in a way tailored to their special properties.

One common challenge to the researchers in these fields is identifying a set of candidate genes that are crucial for the study case, from the thousands of genes in the dataset, that if manipulated can impact the quality and the outcome of the process under study. This candidate set has to be small enough to make the experimental validation of each candidate feasible. One approach is constructing co-expression networks, clustering the genes into modules, usually large ones, then attempting to reduce these modules based on topological feautures [14]. Other approaches, like Short Time-series Expression Miner (STEM), find statistically significant gene patterns and the genes associated with them [15]. Differential gene expression (DEG) analysis is one of the most popular methods to create lists of genes that can be stage-associated. DEG lists provide a good start but often are large in size, and the stage-specific regulators often get diluted in more general genes leading to rather general GO terms when enriched. TFRank [16] is a popular network-based prioritization method, but it doesn't integrate time series expression data. There are other different approaches to prioritize genes and reducing gene lists resulting from previous methods [17], yet none of these specifically take into account the unique properties of cell differentiation.

Another challenge lies in identifying and understanding the important regulatory interactions and programs that trigger and control the expression of different essential genes. One of the most useful and general approaches to address these regulatory programs is via constructing gene regulatory networks (GRNs), typically a directed graph with the genes as nodes and the edges connecting the nodes usually indicating the regulatory interactions. In the past years, many methods and models have been developed to construct GRNs based on either expression data [18], Chip-seq, binding sites analysis, or other data types and models. Some of these models depend solely on one data type to build these networks while others more effectively combine one or more data sources. Despite the general success of some methods which derive GRNs from gene expression data, they have commonly known limitations, such as the inability to deal with time series data in the case of Bayesian Networks (BNs), excessive computational time in the case of Dynamic Baysian Networks (DBNs), and the fact that the number of genes is mostly greater than the number of experimental conditions can cause problems when it comes to methods like Graphical Gaussian Models (GGMs) and BNs [19, 20]. Another different approach is using binding site analysis in the genome to predict the capability of transcription factors (TFs) to regulate the expression of target genes. TFs have the potential to bind to a DNA region via a binding site with a specific pattern of nucleotides that can be recognized by the DNA-binding domain (DBD) associated with each TF. The challenge in this approach lies

mainly in finding the proper library of positional weight matrices (PWMs), the ideal thresholds, and cutoffs and defining the regions of search. The result is an extensive regulatory network that covers a large number of potential regulatory interactions. While these regulatory effects are potentially possible, only a subset of these interactions takes place in a specific context and time. Finding these subsets and refining the global regulatory network according to the biological context under study would result in a more meaningful and case-relevant network.

To tackle these challenges, we constructed a novel workflow and a model of a regulatory network that incorporates the element of time and temporal order, integrates the expression levels of genes, is concise enough to be inspected visually, and identifies candidate regulators efficiently. The method is time and memory efficient, yet it generates a model with a specific architecture to display the primary transcriptional regulators, such as TF genes and miRNAs, and regulatory events unfolding with time. It pre-computes an extensive gene regulatory network that is based on binding site analysis, is independent of the expression data and is used as a background regulatory network. The workflow then uses expression data to identify stage-specific regulators based on their expression pattern. These regulators are finally organized in a cascade architecture that we call a temporal regulatory cascade (TRC). In a TRC, master regulators specific for each stage are organized in ordered vertical columns, and potential regulatory interactions that are based on the background network are displayed as edges between these regulators. To demonstrate this model, we developed an online tool aimed for experimentalists as well as bioinformaticians interested in investigating the regulatory forces that might explain the observed expression of genes in a particular time series dataset. Our novel workflow offers the automatic generation of a TRC from an uploaded time series dataset and visualizes it in an animated interactive manner. In order to facilitate direct interpretation, the results at any stage of the workflow are distilled to an amount that can be handled and analyzed visually, keeping the top significant genes, interactions, and information and discarding those with lower significance and specificity.

In this manuscript, we describe the workflow in detail and report on its application to two time series expression datasets. Both datasets characterize the differentiation of pluripotent stem cells into mature cardiac myocytes and neural progenitors, and the corresponding TRC was generated and analyzed in each case. The main aim was to analyze the specific regulatory activity in each stage, identify and evaluate regulators specific for each time point in the differentiation process, and to test the efficiency of the workflow in re-identifying some well-known case-relevant regulators and regulatory interactions without prior knowledge.

## Materials and methods

### Background regulatory network

A library of position weight matrices (PWMs) from TRANSFAC® [21] is used in combination with the MATCH™ [22] program to predict transcription factor binding sites (TFBSs) in the conserved promoter regions of the human genome as follows.

Based on 49,344 RefSeq-annotated human transcription units (UCSC track refGene, Jan. 22, 2014), the -1kb upstream region was selected as a proximal promoter. The transcription start site (TSS) indicated in RefSeq was used as the reference point.

On the basis of pre-calculated whole genome alignments provided by the UCSC (46_WAY_MULTIZ_hg19) these promoter definitions were utilized to retrieve the sequence conserved regulatory regions between human (hg19), mouse (mm9), dog (canFam2) and cow (bosTau4). Afterwards, gaps resulting from the multiple genome alignment were removed.

MATCH was used to predict potential TFBSs in the previously identified conserved promoter regions, based on all vertebrate defined matrices using the PWM library from TRANS-FAC (release 2013.1, 1446 vertebrate matrices). All matrices with default minFN threshold (minimize false negatives) were used in order to predict potential TFBSs that have at least the quality of an annotated TFBS in TRANSFAC. 1360 out of 1446 TRANSFAC-PWMs had a sequence-conserved TFBS prediction. We ranked all predicted TFBSs associated with each PWM, according to their MATCH score. We chose the best 5% predicted binding sites for each PWM and constructed the background transcriptional regulatory network accordingly. The PWMs are translated to human TF-gene names (HGNC-defined) using the TRANSFAC database. Each TF-gene, identified by its official HGNC-defined gene name, was represented as a node, with a directed edge connecting it with its target gene node. Further information about the construction of the regulatory network can be found in our previous manuscript [23].

The core network included 829 TFs and their 16354 targets, summing up to 749949 interactions. Another expanded network, which includes microRNA binding predictions, was constructed and contained 2239 regulators and 20160 targets. This network was computed once and is independent in the process of its derivation from the expression data, making it usable with every human expression dataset.

While the tool offers the user the option to upload a custom regulatory network to be used for the analysis, we recommend the built-in network just described. The conservation property of these sites makes the prediction ideal for the differentiation context since several pieces of research have shown that conserved regions in the DNA are critical binding sites for development and differentiation [24–27].

## Temporal regulatory cascades (TRCs)

The method utilizes the concept of constructing template expression patterns that represent an expression behaviour of interest, then attracting genes that behave similarly to these patterns using correlation. The template patterns we used were stage-specific patterns, peaking at one time point only, and denoted by template peak patterns (TPPs). While different kinds of template patterns can be used, we chose the single-peak TPPs, as a default for its ability to attract stage-specific regulators that are unique to each time point. Regulatory interactions are queried from the background regulatory network and form the edges between the genes in the cascade accordingly.

**TRC construction steps.**

**Step 1**: Create a library of TPPs, one TPP for each time point in the dataset. For each time point the corresponding TPP has an expression level of 100 percent at that time point and zero every other time point(Fig 1A).

**Step 2**: For each TPP, calculate the top $n$ correlated regulators to this reference pattern (Fig 1B). These genes are said to be the stage-specific regulators of stage $s$ and are displayed in the same column (Fig 2). If a time point has no correlated regulators, no column is created for this stage in the TRC.

**Step 3**: All regulatory interactions between the regulators of the same stage are mapped, according to their connections in the background regulatory network, and represented in the form of directed edges.

**Step 4**: All regulatory interactions between the regulators of stage $s$ and the next stage are mapped according to their connections in the background regulatory network and

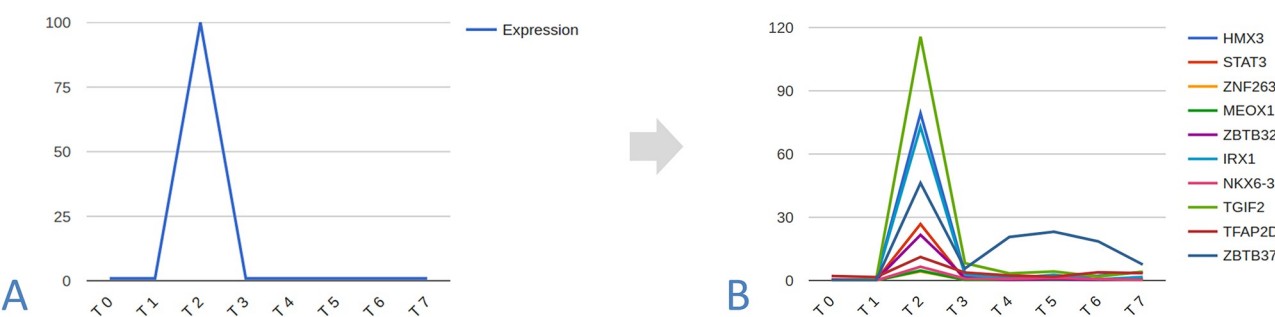

**Fig 1. Indetifying stage-specific regulators.** (A) The TPP of T2: The template peaking pattern is calculated where the expression is at 100 percent T2 and zero every other timepoint. One TPP for each time point is calculated similarly and the collection of these TPPs form the TPP library. (B) Top 10 regulators that are highly correlated with the previous TPP of T2 and their noticeable T2-specific peaking pattern, these regulators will form nodes in the T2 column in the TRC, the same is done for every TPP in the library.

represented by directed edges, linking each stage to the next and tying the cascade together (Fig 2).

**Parameters.**   To adjust the temporal regulatory cascade, we use three primary parameters:

*minE*: A threshold for gene expression levels. A gene that does not have an expression level higher than this threshold in any of the replicates or time points is eliminated and omitted from the calculation that leads to the TRC. This eliminates peaking genes that are lowly expressed even at their peak.

*minC*: A minimum correlation threshold. Regulatory genes that have a correlation above this threshold to the TPP of a stage are kept as the initial set of regulators associated with that stage.

*maxS*: The maximum number of genes that can be associated with a specific time point. The initial regulators associated with a time point based on *minC* are sorted by their correlation to the TPP of that stage, and the top *n* (*maxS*) regulators are picked to be in the column associated with the stage. If the initial regulators set has less than *maxS* genes, then the whole set is taken. The max number of nodes in the cascade is *maxS* multiplied by the number of time points.

## Implementation

This workflow was implemented in the form of a web service with an interactive visual web interface, which eliminates then the need to install any additional software. The algorithm to generate the TRCs was implemented using Java. In order to display the resulting TRC, a visualizer was implemented using JavaScript, utilizing, and extending the Cytoscape library cy.js. The framework used PHP to manage the files and sessions. The visualizer was embedded in an interactive webpage that includes helpful information such as graphs of the expression levels of the genes in the cascade, tables, and metrics, in addition to direct links to perform GO enrichment and other workflows in the platform. The web tool is a part of a more comprehensive web service that revolves around gene regulation and expression data analysis that is under construction.

## Data

While any formatted time series data can be used as input, this model performs the best with RNA-Seq data over other sources of inferior quality and less variability such as microarray

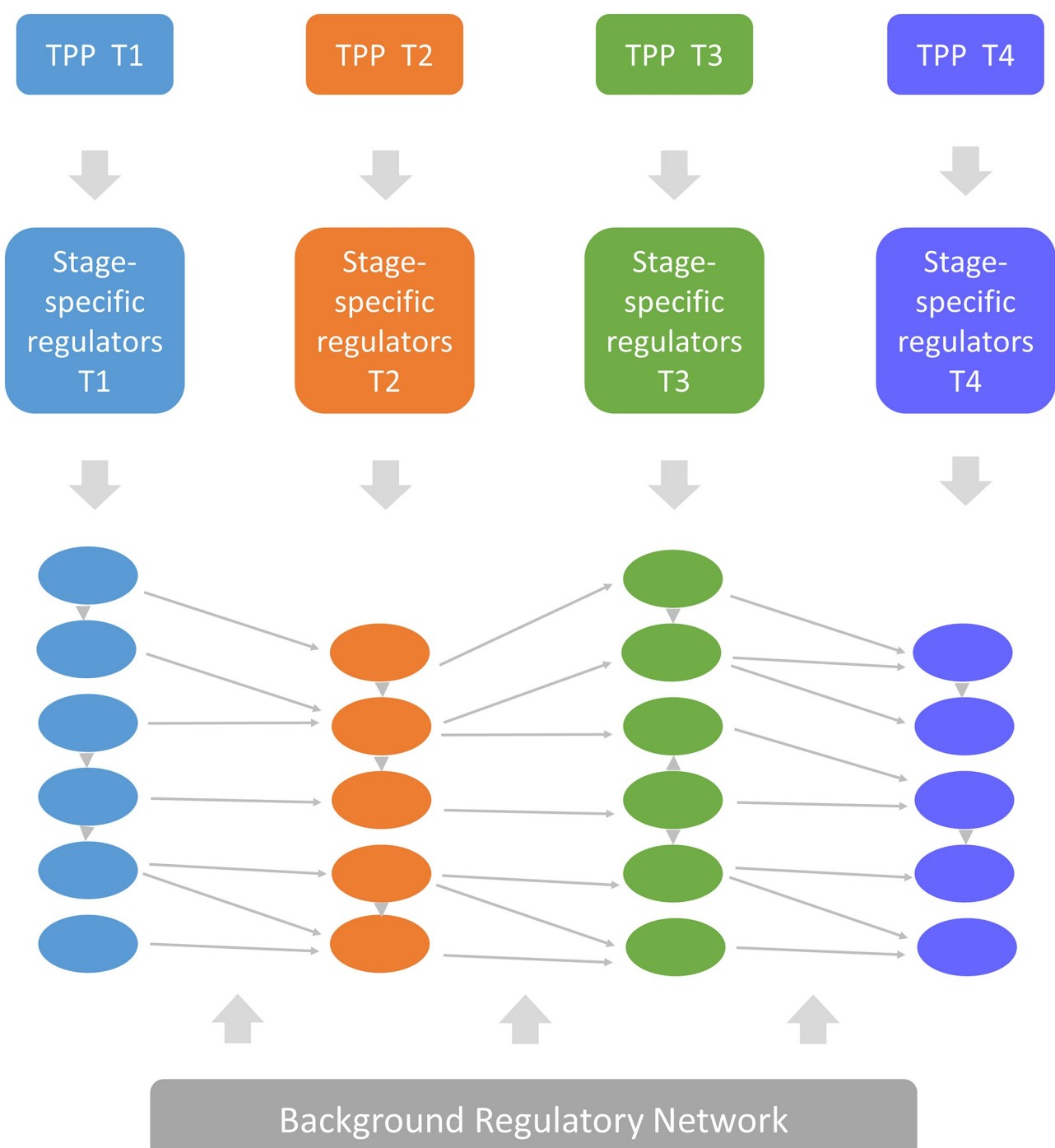

**Fig 2. The TRC workflow.** Regulators specific for each time point are grouped in the same column with the same color and sorted by their correlation to the TPP of that stage. The edges between the two stages and within the same stage are retrieved and mapped from the regulatory network.

data. Normalized input data provides a better quality TRC, nevertheless even using the raw counts leads to reasonably significant TRCs. As study cases to demonstrate the TRC model, two time series gene expression datasets were used and denoted Dataset1 and Dataset2.

Dataset1 was assembled using public RNA-Seq data that is captured during the differentiation of H1 derived human neuronal precursor cells (NPCs) across the days 0,1,2,4,5,11, and 18 after induction of neuronal differentiation. Publicly available DEG and GO enrichment analysis on the same dataset was used for comparison. The dataset and the analysis results could be found in the expression Atlas under the accession E-GEOD-56785. The assembled and formatted data can be found in S1 File.

Dataset2 was derived from the normalized expression datasets from the previously published study by Qing Liu et al [28], publicly available in the GEO repository under the accession number GSE85332. We chose one of the four expression datasets available, the RNA-Seq profiling of the differentiation of C20 derived cardiomyocytes at four stages: pluripotent stem cells (day 0), mesoderm (day 2), cardiac mesoderm (day 4), and differentiated cardiomyocytes (day 30). The assembled and formatted data can be found in S2 File.

## GO enrichment

To evaluate the relevance of the gene sets in each stage, Gene Ontology (GO) enrichment analysis using the biological processes and a Fisher's Exact test on each column in these cascades was applied using one set at a time as an input. Terms that have a pvalue less than 0.05 after the Bonferroni correction are sorted by their fold enrichment and the top terms were examined. These terms were evaluated based on their consistency with the differentiation stage under observation at that time point.

## Results

We applied the TRC workflow to Dataset1 and Dataset2 and generated a cascade for each study case. In addition to the GO enrichment, detailed literature research was performed, investigating the roles of the different regulators predicted by the cascade.

### Neural differentiation cascade

Upon the visual inspection of the cascade, we observe a missing time point that is day 2, indicating that this time point does not have any peak strength or any genes that exceed the correlation threshold to the TPP, suggesting that day 2 might be a time point that doesn't underly any unique stage-specific activity (Fig 3).

Examining the GO enrichment of each time point reveals high enrichment of relevant terms in day 1 and day 11. Regulators of day 1 showed enrichment for specific terms such as cell and neuron fate commitment, neuron differentiation, and cell differentiation in the spinal cord. Regulators of day 11 showed high enrichment of even more specific terms such as spinal cord association neuron differentiation, dorsal spinal cord development, cell fate determination, cell differentiation in the spinal cord, hindbrain development. On the other hand, examining the GO enrichment based on the DEG analysis publicly available for the same dataset, differentially expressed genes in day 0 vs. day 1 and day 0 vs. day 11 showed no significant enrichment of specific terms associated with neural development but rather more general terms.

A deeper look into the identity of the regulators in the cascade shows that *OLIG1* and *OLIG3*, which are known for their importance in neural and spinal development [29–31], are active in day 1, suggesting that their importance lies in the earlier part of the differentiation. A microRNA *MIR3659* peaking at day 1 with a high indegree raises the question on the nature of its involvement in neural differentiation, which needs to be further investigated. *PAX2* on day 11, with the highest outdegree, regulates 13 different regulators in the same and next time point which hints that its known essential role in neural development [32–34] is due to its

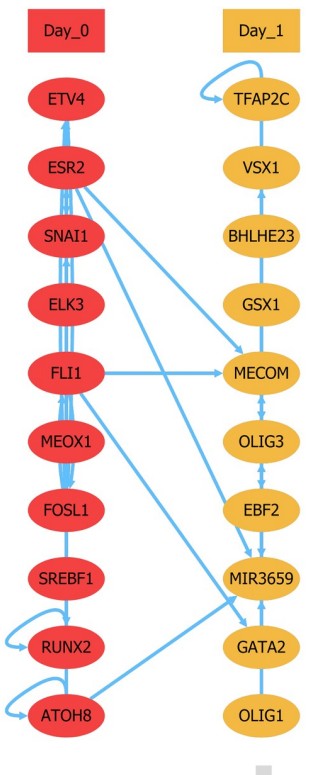

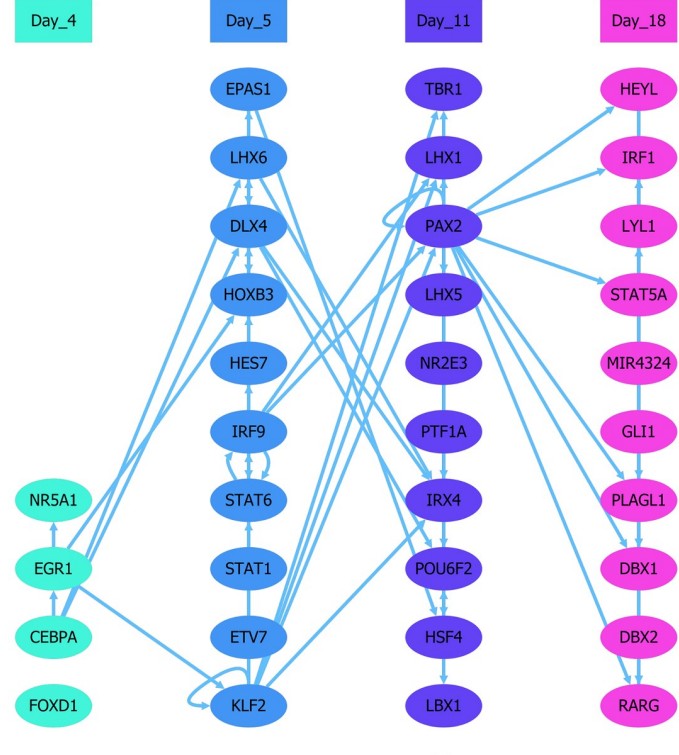

**Fig 3. Neural differentiation cascade.** The TRC generated for the differentiation of neural proginators based on dataset 1 and the following parameters: $minC = 0.6$, $minE = 4$, and $maxS = 10$.

regulatory impact on a big set of neural regulators. *KLF2* in day 5 stands out as a potential significant regulator of the day 11 regulatory wave due to its potential ability to regulate a big portion of day 11 regulators. The TRC shows an overall same-stage presence of certain TFs that belong to the same family or subfamily according to the classification experimental conditions TFs in TFClass [35, 36], such as *OLIG1*, *OLIG3* and *BHLHE23* in day 1, *STAT1* and *STAT6* in day 5, the *LHX1* and *LHX5* in day 11, *DBX1* and *DBX2* in day 18. A hypothesis can be made that these TFs are part of the redundancy that leads to the robustness of such regulatory programs, or that these families and subfamilies of TFs collaborate in certain regulatory stages.

## Cardiac differentiation cascade

Regulators of the first time point show enrichment of terms related to stem cell maintenance, which is coherent with the biological context since the process of differentiation has not started yet, and the cells are still in the induced stem cell state (Fig 4). These regulators could be

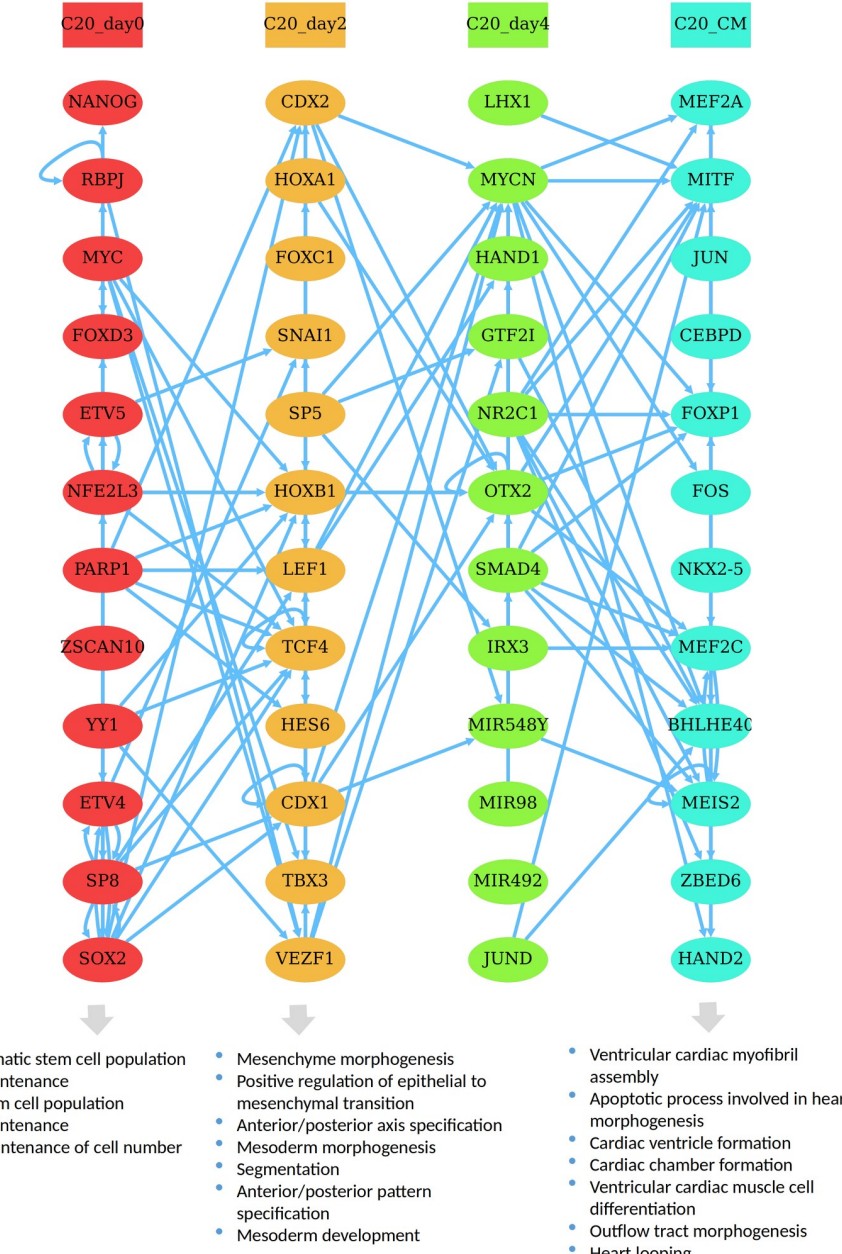

**Fig 4. Cardiac differentiation cascade.** The TRC generated for the differentiation of cardiomyocytes based on dataset2 and the following parameters: $minC = 0.6$, $minE = 30$, and $maxS = 12$.

essential for maintaining the pluripotency state and also could be repressing differentiation. Regulators of day 2 show enrichment of terms associated with mesenchymal and mesoderm morphogenesis, which give rise to cardiac cells. Regulators of the last stage the cardiomyocyte (CM) stage show high enrichment of heart-specific terms such as cardiac ventricle and chamber formation, ventricular cardiac muscle differentiation, heart looping, and outflow tract morphogenesis. These terms show a high consistency with the underlying stage of differentiation reported by the experiment.

In the first time point TFs associated with maintanining the pluripotency state like *NANOG* [37], *PARP1* [38], *SOX2* [39], *MYC* [40, 41], *ETV4* and *ETV5* [42] appear. *CDX1* and *CDX2* [43] which are known to modulate early cardiogenesis peak at day 2, alongside some potentially important early cardiac regulators such as *TCF4* and *LEF1*. On day 4, *MYCN* stands out with a high outdegree and indegree confirming its known role in heart development [44] along side with some potential candidate regulators such as *LHX1*, *OTX2*, *NR2C1*, *MIR548Y*. The last stage where the cardiomyocytes have already matured, features core regulators essential for cardiac development such as *MEF2C* [45], *HAND2* [46], *NKX2-5* [47], *MEIS2* [48], *MITF* [49], FOXP1 [50] and some new candidate regulators that could be significant in the cardiac maturation such as *MEF2A* and *BHLHE40*. Like in the previous dataset, a strong same-stage presence of certain TF family members is observed, such as the members of the HOX family CDX1, CDX2, HOXA1 and HOXB1 in day 2.

## Discussion

Unlike some of the classic regulatory models such as BNs, the TRC model takes advantage of the sequential order of the time series data to allow more intricate interpretations of regulatory interactions. It takes advantage of the emerging property from the peaking patterns, that is: each node in the cascade is positively correlated in its expression pattern to the other nodes in the same stage (Fig 5A), and correlated via a time-lagged correlation to the nodes in the other stages (Fig 5C). Thus each edge in the cascade is always coupled with a correlation between the expression pattern of the regulator and its target. This coupling can be viewed as a reinforcement of the regulatory interaction predictive quality and gives it an edge over interactions based solely on the binding site analysis or solely derived from gene expression data. From another view, the binding site prediction behind the edge can explain the perceived correlation in the expression patterns between the target and the source. Fig 5 summarizes the five common types of regulatory interactions displayed within the cascade through edge patterns. Fig 5A is an example of a regulatory interaction coupled with high positive correlation indicating that X is potentially one of the activators of Y and contributes to its peaking pattern. Y is inactive where X is inactive and activated when X is activated (stage i), coupled with the fact that X can bind to the promoter of Y, this hypothesis of the regulatory influence of X on Y is strongly enforced. Fig 5A has a one-direction property that supports the causality, whereas cases such as the double edge displayed in Fig 5B cannot decisively assert whether X is an activator of Y or the other way around due to the non-causal nature of correlation and the double potential of these regulators to bind to each other's promoters. Fig 5C is an example of where a regulator in a certain stage potentially needs more time to activate the target thus the target is activated after a time delay and captured in the next stage. This kind of regulatory behavior has been shown and captured using time-lagged correlation models. Another common hypothesis that surrounds co-expressed genes is that they might be coregulated by a master regulator or a set of master regulators. Some of these master regulators can be captured through configurations in the cascade where a regulator emerging in a stage single-handedly has the potential to activate a wide set of correlated regulators, whether in the same stage as in the case of Fig 5D or a set of targets in the next stage via a time-lagged regulation as shown in Fig 5E.

The previous analyses of the two datasets showed clear stage-specific regulatory waves and a GO enrichment that is highly consistent with the biological context of the experiment and, even more specifically, the context in the particular time points of the experiment. The question arises whether these peaking profiles and case-specific GO enrichments are statistically significant, and constitute a characteristic of developmental gene expression datasets in particular, or if they randomly occur in any dataset. While applying the TRC model to a sufficiently

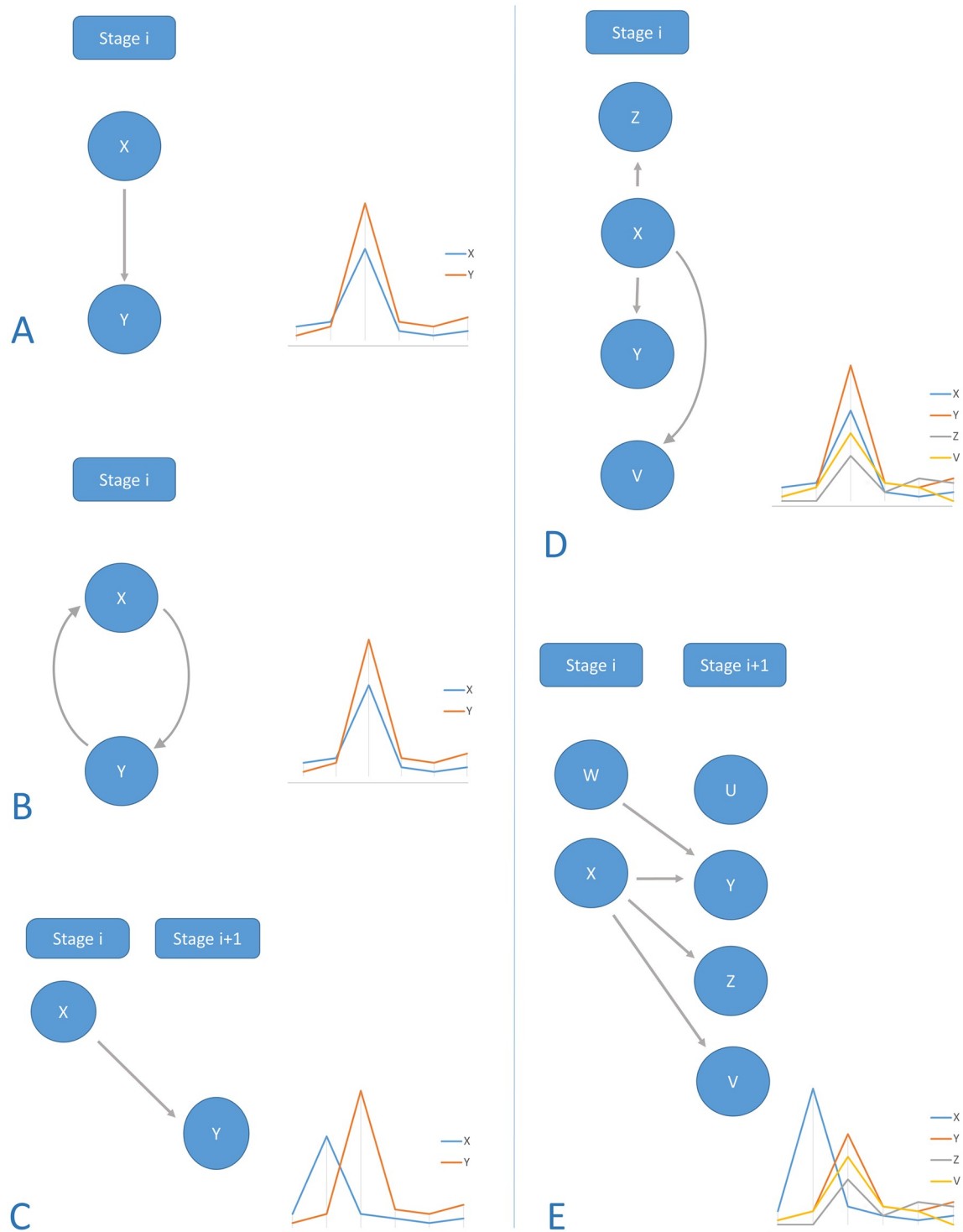

**Fig 5. Different cases of regulatory interactions contained in the TRC model.** (A) A one-way regulatory prediction within one stage coupled with a high positive correlation. (B) A two-way regulatory prediction within one stage coupled with a high positive correlation. (C) A regulatory interaction from one stage to the next, coupled with a high positive time-lagged correlation. (D) X a potential master regulator of X, Y, and Z coupled with a high positive correlation to each of its targets. (E) X a potential master regulator activating X, Y and Z coupled with a high positive time-lagged correlation to each of its targets.

large number of random and shuffled datasets and evaluating the resulting TRCs would be optimal to proof the statistical significance of the results, it is merely unfeasible due to the manual process of assessing the resulting TRCs. Alternatively, we applied the model to randomly generated and shuffled gene expression datasets (see the supplementary files) aiming towards a comparative analysis rather than a statistical proof of significance. We examined the resulting TRCs in terms of the GO enrichment of the stages to evaluate their relevance compared to a TRC generated from a real experimental dataset. The first test involved shuffling dataset2 by re-assigning genes to other expression profiles (S3 File), to check whether any set of peaking regulators will show a specific GO enrichment, and none of the stages did lead to any relevant terms. The test was repeated by shuffling the regulator's profiles only, and the enrichment was again insignificant. The previous test showed that the identity of the peaking genes is essential, precise, and specific. Moreover, the workflow was applied to dataset2 without restricting the stage-specific sets to regulators only. Interestingly, the generated cascade was overwhelmed by non-regulatory genes and the GO enrichment showed no significant terms in any of the stages, with the exception of two terms related to cardiac muscle differentiation in the last stage (S1 Fig). This observation supports the choice in the TRC model of limiting the cascade to regulators where less relevant non-regulatory genes do not dilute the small stage-specific gene sets. Next, dataset2 was shuffled by permuting all the values in the expression matrix (S4 File). The result was again a lack of significance in GO the enrichment terms. The last test was applying the TRC workflow to a randomly generated gene expression dataset, using the gene names and the time points from dataset2 combined with randomly generated expression values (S5 File). The GO enrichment showed the absence of any relevant significant terms again.

The default library used in this model is the one-stage peak pattern library, which works optimally with development and differentiation. However this library can be changed, and multiple libraries for different biological contexts such as diseases and immune responses can be developed accordingly, which would require further research or alternatively allowing the user to construct a custom library in the future.

One drawback of this model is the fact that it does not capture every important regulator, particularly those regulators that are expressed in multiple consecutive or non-consecutive time points. However, we argue that the sets of regulators identified by the cascade contain a large percentage of essential stage-specific regulators which is supported by the GO enrichment. On the other hand, the regulatory network might not cover every TF due to missing PWM information or lack of conservation. Another more general drawback is the fact that the model relies on transcript levels which do not translate directly into protein levels, but relative measures [51] [52] can be a potential method for further analysis whenever protein data is not available. Moreover, the candidate regulators can provide a small concise set for a proteomic investigation as a next step in the experiment. The captured regulators can also provide a starting point for further analysis such as target set enrichment analysis, pathway analysis, and investigating the potential collaboration of regulators using tools such as PC-Traff [53]. The TRC model merely lays down, in place, some important starting pieces that can be built on to complete the biological puzzle of developmental regulatory programs.

The unique type of the output of the TRC makes it difficult to accurately compare it to other existing methods, as no other method has the same definition of a regulatory cascade. However, we utilized the context-relevance of the GO enrichment of the gene sets predicted by other methods as a basis for the comparison. We first applied the STEM in order to predict the top 10 significant gene expression patterns in the cardiac differentiation dataset and evaluated the GO enrichment of the genes set associated with each of these profiles. The GO enrichment of these sets showed very general terms not specific to the cardiac differentiation context.

Next, we applied iDREM [54], which we consider the closest method to the TRC in terms of inputs and aims, using the cardiac differentiation dataset and the regulatory network provided by iDREM (human_predicted_1000), to generate a dynamic regulatory network. The resulting model was in the form of a dynamic regulatory map that highlights major bifurcation events, each of which has a list of associated regulatory genes. The GO enrichment of these gene lists showed a mild enrichment of developmental GO terms in some bifurcation points and no enrichment in most of the others. However, proving the validity of a generated network or cascade requires an actual experimental validation of the predicted regulatory interactions in the particular cellular context, which is currently unpractical.

This workflow is built within a broader framework dedicated to studying regulation from different points of view. It blends expression data and a regulatory network and links concepts such as coexpression and coregulation forming a more extensive tool. Users can interactively investigate different hypothesis and track different genes and regulators of interest exploring the regulatory forces governing the time series data, the timing of such forces and the impact of such regulatory interactions on the expression of genes and regulators.

## Conclusion

We developed a workflow to analyze and represent regulatory cascades and a web tool based on the corresponding model. It takes time series expression data as an input, generates and visualizes an interactive cascade that identifies relevant and stage-specific regulators associated with each time point and the interactions between these regulators. The workflow was applied to multiple datasets that revolved around cell differentiation and was successful in identifying previously-known TFs relevant to the time points and the cell types, in addition to some new candidate regulators, as well as pinpointing the time points were unique regulation activities are emerging. A demo of the web tool is available under TF-investigator.sybig.de/TRC.

## Supporting information

**S1 File. NPC differentiation (Dataset1).** The formatted data expression file based on human H1-derived NPC differentiation differentiation. This format is ready for upload via the webtool.
(CSV)

**S2 File. Cardiac differentiation (Dataset2).** The formatted data expression file based on C20 derived cardiomyocyte differentiation. This format is ready for upload via the webtool.
(CSV)

**S3 File. Shuffled profile assignment of dataset2.** A version of dataset2 where gene profiles are randomly re-assigned. This format is ready for upload via the webtool.
(CSV)

**S4 File. Shuffled dataset2 by permuting the matrix.** A version of dataset2 where cells in the expression matrix are permuted across columns and rows. This format is ready for upload via the webtool.
(CSV)

**S5 File. Random expression values with dataset2 time points and gene names.** Random expression values with time points and gene names taken from dataset2. This format is ready for upload via the webtool.
(CSV)

**S1 Fig. TRC based on dataset2 where regulatory and non-regulatory genes are included.**
Stage-specific gene sets are not restricted to regulators in this example. This allows the TRC to
include peaking non regulatory genes as well.
(TIF)

# Acknowledgments

We would like to thank Sebastian Zeidler and Hryhorii Chereda for their insights and helpful
discussions.

# Author Contributions

**Conceptualization:** Rayan Daou, Edgar Wingender, Mehmet Gültas, Martin Haubrock.

**Methodology:** Rayan Daou.

**Software:** Rayan Daou.

**Supervision:** Tim Beißbarth, Edgar Wingender, Martin Haubrock.

**Visualization:** Rayan Daou.

**Writing – original draft:** Rayan Daou.

**Writing – review & editing:** Tim Beißbarth, Edgar Wingender, Mehmet Gültas, Martin
Haubrock.

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
