## [Decision Letter · Decision Letter 0]

2 Jan 2020

PONE-D-19-28215

Constructing temporal regulatory cascades in the context of development and cell differentiation

PLOS ONE

Dear Dr. Daou,

Thank you for submitting your manuscript to PLOS ONE. After careful consideration, we feel that it has merit but does not fully meet PLOS ONE’s publication criteria as it currently stands. Therefore, we invite you to submit a revised version of the manuscript that addresses the points raised during the review process.

We would appreciate receiving your revised manuscript by Feb 16 2020 11:59PM. To enhance the reproducibility of your results, we recommend that if applicable you deposit your laboratory protocols in protocols.io, where a protocol can be assigned its own identifier (DOI) such that it can be cited independently in the future. For instructions see: http://journals.plos.org/plosone/s/submission-guidelines#loc-laboratory-protocols

We look forward to receiving your revised manuscript.

Kind regards,

Roberto Mantovani

Academic Editor

PLOS ONE

Journal Requirements:

2. Please upload a new copy of Figure 1 as the detail is not clear.

Please follow the link for more information: http://blogs.PLOS.org/everyone/2011/05/10/how-to-check-your-manuscript-image-quality-in-editorial-manager/

Reviewers' comments:

Reviewer's Responses to Questions

**Comments to the Author**

1. Is the manuscript technically sound, and do the data support the conclusions?

Reviewer #1: Yes

2. Has the statistical analysis been performed appropriately and rigorously? 

Reviewer #1: N/A

3. Have the authors made all data underlying the findings in their manuscript fully available?

Reviewer #1: Yes

4. Is the manuscript presented in an intelligible fashion and written in standard English?

Reviewer #1: Yes

5. Review Comments to the Author

Reviewer #1: The authors propose a novel approach to explore time-series gene expression data, integrating them with the underlying regulatory network. The idea of identifying time-point-specific regulators candidates from the expression profiles and then mapping them back to a background regulatory network in order to figure out temporal relationships and identify a small set of key regulators for each time-point is surely worth to be investigated. The two examples reported by the authors are also interesting, but perhaps a more comprehensive selection of use cases could add strength to the manuscript. A comparison with other tools to investigate temporal regulatory cascades would also be interesting. Unluckily, the implementation of the user front-end to the method and of the documentation seems still in a very preliminary phase, and this needs to be addressed before publication. Also, the methods used to draw the regulatory network could be somewhat refined.

Major:

- The design of the gene regulatory network is of fundamental importance for this method, but this aspect seems to be a bit overlooked in the manuscript. In particular additional discussion and implementation efforts should be devoted to:

--how is a promoter defined?

--Why only are Transfac PWMs used when there exist other libraries like Jaspar, with the additional advantage that they are not commercial?

--The PWM score threshold method to associate a TF to a given promoter is very blunt and could be refined. In any case, how is the threshold chosen? A simple reference to another paper is not sufficient for a matter of this importance for this method.

-- Why selecting mammals specific conserved regions (human, mouse, dog, and cow) genomes works better than regions conserved among vertebrates instead? Would it be possible, for example, to identify mammal-specific regulators comparing the results obtained with networks built using vertebrate conserved regulatory regions instead of mammals ones?

-- On the other hand, would it be possible to rely on chromatin states instead of conservation in order to identify functional promoter regions and how the results would change?

- A comparison of results with other methods for temporal gene networks adopting other approaches could help readers and potential users in understanding the advantages of this method. For example, the cascade R package [Jung et al.] could be used as a benchmark.

- The Neural and Cardyomyocytes examples reported by the authors are suggestive, but a somewhat more extensive selection of examples could be helpful. In particular, it would be interesting to see if the method can be applied with success also to non-human time series. For example, an interesting dataset could be the time-resolved transcriptome of C. elegans [Boeck et al.]. PWMs for C. elegans are also available in Jaspar.

- This is the weakest point of the manuscript. The implementation of the method is not usable in its current state. At the link that has been provided, there is a very blunt interface without any documentation. No information, for example, is provided on the format of the input file, no license, no terms of use, no tutorial or explanation on how to use it, and understand the output. If this is open software, it should be made available using standard repositories. There is no way to select the background regulatory network to be used, so it seems to works only for human data using the default background regulatory network built by the authors, but it would be much better to provide more topologies to users and maybe also let users provide their topologies.

6. PLOS authors have the option to publish the peer review history of their article (what does this mean?). If published, this will include your full peer review and any attached files.

Reviewer #1: No

---

## [Author Response · Author response to Decision Letter 0]

18 Feb 2020

Responses to editor’s comments:

1. Please ensure that your manuscript meets PLOS ONE's style requirements, including those for file naming. The PLOS ONE style templates can be found at:

http://www.journals.plos.org/plosone/s/file?id=wjVg/PLOSOne_formatting_sample_main_body.pdf
http://www.journals.plos.org/plosone/s/file?id=ba62/PLOSOne_formatting_sample_title_authors_affiliations.pdf

We adjusted the names of the supplementary data files from “S.. Dataset” to “S.. File” and “S1 Figure” to “S1 Fig.”. We hope that it now fits the style requirements, as described in the referred templates.

2. Please upload a new copy of Figure 1 as the detail is not clear.

As pointed, Figure 1 was not clear, mainly because of the amount of information that rendered the details hard to read. So we decided to substitute the figure with two new separate figures Figure 1 and Figure 2. The manuscript was adjusted accordingly.

 

Responses to reviewer #1:

The design of the gene regulatory network is of fundamental importance for this method, but this aspect seems to be a bit overlooked in the manuscript.

We agree with the reviewer on the importance of the design of the gene regulatory network (GRN) used. However, since the focus of the present paper was on the construction of regulatory time cascades, which can be done on the basis of any GRN, we intentionally cut this aspect short. Including all technical details of its construction would also require more detailed considerations of data sources used, algorithms used for the detection of transcription factor binding sites, thresholds applied, etc. The network we constructed, described previously and used in a number of studies consistently proved functional. Its construction was a considerable effort, and to re-do this with another data source would preclude a timely publication of the results described in our paper without adding to its main points. As said, the method described can be applied using any other GRN; however, as suggested by the reviewer, we added the possibility for users to exchange the precomputed GRN by a customized one in case the user has one at hand . Of course, the results may differ. We expanded the background regulatory network section in the materials and methods to include the details of the construction of the network.

To address the questions raised by Reviewer #1 in more detail:

--how is a promoter defined?

Based on 49,344 RefSeq-annotated human transcription units (UCSC track refGene, Jan. 22, 2014), the -1kb upstream region was selected as a proximal promoter. The transcription start site (TSS) indicated in RefSeq was used as the reference point.

On the basis of pre-calculated whole-genome alignments provided by the UCSC (46_ WAY_MULTIZ_hg19) these promoter definitions were utilized to retrieve the sequence conserved regulatory regions between human (hg19), mouse (mm9), dog (canFam2) and cow (bosTau4). Afterward, gaps resulting from the multiple genome alignment were removed.

--Why only are Transfac PWMs used when there exist other libraries like Jaspar, with the additional advantage that they are not commercial?

TRANSFAC has been the first database about gene regulatory components and binding site models (PWMs) and still is the most comprehensive data source for these entities, also comprising the Jaspar collection.

--The PWM score threshold method to associate a TF to a given promoter is very blunt and could be refined. In any case, how is the threshold chosen? A simple reference to another paper is not sufficient for a matter of this importance for this method.

MATCH was used to predict potential TFBSs in the previously identified conserved promoter regions, based on all vertebrate defined matrices using the PWM library from TRANSFAC (release 2013.1, 1446 vertebrate matrices) . All matrices with default minFN threshold (minimize false negatives) were used in order to predict potential TFBSs that have at least the quality of an annotated TFBS in TRANSFAC. 1360 out of 1446 TRANSFAC-PWMs had a sequence-conserved TFBS prediction. We ranked all predicted TFBSs associated with each PWM, according to their MATCH score. We chose the best 5% predicted binding sites for each PWM and constructed the background transcriptional regulatory network accordingly. The PWMs are translated to human TF-gene names (HGNC-defined) using the TRANSFAC database. Each TF-gene, identified by its official HGNC-defined gene name, was represented as a node, with a directed edge connecting it with its target gene node.

-- Why selecting mammals specific conserved regions (human, mouse, dog, and cow) genomes works better than regions conserved among vertebrates instead? Would it be possible, for example, to identify mammal-specific regulators comparing the results obtained with networks built using vertebrate conserved regulatory regions instead of mammals ones?

This is certainly a valid suggestion by Reviewer #1, but identifying mammal-specific regulators was not in the scope of our study. When extending conservation to non-mammalian genomes, the number of conserved binding sites would drop considerably and would not help in interpreting, as in this study, human data. On the other side, including monkey or rat would not add since they are too close to human and mouse, resp.

-- On the other hand, would it be possible to rely on chromatin states instead of conservation in order to identify functional promoter regions and how the results would change?

Since chromatin states are highly dependent on the cellular context, corresponding data would always refer to individual cells / cell lines only and would not help in constructing a comprehensive GRN.

- A comparison of results with other methods for temporal gene networks adopting other approaches could help readers and potential users in understanding the advantages of this method. For example, the cascade R package [Jung et al.] could be used as a benchmark.

The unique type of the output of the TRC makes it difficult to accurately compare it to other existing methods, as no other method has the same definition of a regulatory cascade. However, we utilized the context-relevance of the GO enrichment of the gene sets predicted by other methods as a benchmark for the comparison. We considered the suggested Cascade R package for the comparison however we felt it was not properly maintained and subsequently, we couldn't manage to run it on the required dataset, and more importantly we needed a method that also combines some sort of a precomputed regulatory network to make the comparison fair. To compare our suggested template profiles to other profile possibilities, we first applied the STEM in order to predict the top 10 significant gene expression patterns in the cardiac differentiation dataset and evaluated the GO enrichment of the genes set associated with each of these profiles.The GO enrichment of these sets showed very general terms not specific to the cardiac differentiation context. Next, we applied iDREM, which we consider the closest method to the TRC in terms of inputs and aims, using the cardiac differentiation dataset and the regulatory network provided by iDREM (human¬ predicted¬1000), to generate a dynamic regulatory network. The resulting model was in the form of a dynamic regulatory map that highlights major bifurcation events, each of which has a list of associated regulatory genes. The GO enrichment of these gene lists showed a mild enrichment of developmental GO terms in some bifurcation points and no enrichment in most of the others. However, proving the validity of a generated network or cascade requires an actual experimental validation of the predicted regulatory interactions in the particular cellular context, which is currently unpractical. We added a paragraph in the discussion section of the manuscript, where we address the comparison part just discussed.

- The Neural and Cardyomyocytes examples reported by the authors are suggestive, but a somewhat more extensive selection of examples could be helpful. In particular, it would be interesting to see if the method can be applied with success also to non-human time series. For example, an interesting dataset could be the time-resolved transcriptome of C. elegans [Boeck et al.]. PWMs for C. elegans are also available in Jaspar.

The GRN constructed was specifically designed to support the reliable interpretation of mammalian, in particular human gene expression time series for biomedical research. Analyzing non-mammalian data would require a different GRN, as well as specific expertise and field of focus related to that species. We chose those examples where the validation of the results on the basis of existing knowledge was possible. However, with the added option of uploading their own regulatory network, the users can now explore different expression datasets and networks related to other species.

- This is the weakest point of the manuscript. The implementation of the method is not usable in its current state. At the link that has been provided, there is a very blunt interface without any documentation. No information, for example, is provided on the format of the input file, no license, no terms of use, no tutorial or explanation on how to use it, and understand the output. If this is open software, it should be made available using standard repositories. There is no way to select the background regulatory network to be used, so it seems to works only for human data using the default background regulatory network built by the authors, but it would be much better to provide more topologies to users and maybe also let users provide their topologies.

We agree with the reviewer’s points, and thus we adjusted the implementation of the webtool accordingly. We added a manual that explains, for example, the file formats and the underlying methods, workflows, and parameters. We also added a tutorial section as well as help buttons and icons that are intended to guide the user through webtool. And most importantly, we added the option that allows the user to upload his own regulatory network as well and use it as a background network for the analysis. 

We also added supportive workflows, which we had in mind from the beginning; these workflows such as the co-expression analysis are not novel in their methodology and are based on classic methods. Thus we did not go into the description of these workflows in the paper, keeping the focus in the novel TRC method; however, in the manual , a detailed description of these workflows can be found. This creates an exploratory platform where a user can further explore different aspects of regulation and gene expression analysis. For this manuscript, the relevant workflow would be the “TRC analysis” workflow which can be found as the first option in the workflows page that appears after the user uploads his data or uses the sample data or built-in network. After choosing the TRC analysis workflow, the user is forwarded to a page where the parameters are adjusted accordingly.

Thank you again for your time and effort,

Best regards

Rayan Daou

---

## [Decision Letter · Decision Letter 1]

23 Mar 2020

Constructing temporal regulatory cascades in the context of development and cell differentiation

PONE-D-19-28215R1

Dear Dr. Daou,

We are pleased to inform you that your manuscript has been judged scientifically suitable for publication and will be formally accepted for publication once it complies with all outstanding technical requirements.

With kind regards,

Roberto Mantovani

Academic Editor

PLOS ONE

Additional Editor Comments (optional):

Reviewers' comments:

Reviewer's Responses to Questions

**Comments to the Author**

1. If the authors have adequately addressed your comments raised in a previous round of review and you feel that this manuscript is now acceptable for publication, you may indicate that here to bypass the “Comments to the Author” section, enter your conflict of interest statement in the “Confidential to Editor” section, and submit your "Accept" recommendation.

Reviewer #1: All comments have been addressed

2. Is the manuscript technically sound, and do the data support the conclusions?

Reviewer #1: Yes

3. Has the statistical analysis been performed appropriately and rigorously? 

Reviewer #1: Yes

4. Have the authors made all data underlying the findings in their manuscript fully available?

Reviewer #1: Yes

5. Is the manuscript presented in an intelligible fashion and written in standard English?

Reviewer #1: Yes

6. Review Comments to the Author

Reviewer #1: (No Response)

7. PLOS authors have the option to publish the peer review history of their article (what does this mean?). If published, this will include your full peer review and any attached files.

Reviewer #1: No

---

## [Editor Report · Acceptance letter]

26 Mar 2020

PONE-D-19-28215R1 

Constructing temporal regulatory cascades in the context of development and cell differentiation 

Dear Dr. Daou:

I am pleased to inform you that your manuscript has been deemed suitable for publication in PLOS ONE. Congratulations! Your manuscript is now with our production department. 

With kind regards,

on behalf of

Prof. Roberto Mantovani 

Academic Editor

PLOS ONE